# DeepSight: Long-Horizon World Modeling via Latent States Prediction for End-to-End Autonomous Driving

**Lingjun Zhang** [* 1 2]  **Changjie Wu** [* 2]  **Linzhe Shi** [2]  **Jiangyang Li** [2 3]  **Jiaxin Liu** [1]  **Lei Yang** [4]  **Hang Zhang** [2]
**Mu Xu** [2]  **Hong Wang** [1]

## Abstract

End-to-end autonomous driving systems are increasingly integrating Vision-Language Model (VLM) architectures, incorporating text reasoning or visual reasoning to enhance the robustness and accuracy of driving decisions. However, the reasoning mechanisms employed in most methods are direct adaptations from general domains, lacking in-depth exploration tailored to autonomous driving scenarios, particularly within visual reasoning modules. In this paper, we propose a driving world model that performs parallel prediction of latent semantic features for consecutive future frames in the bird's-eye-view (BEV) space, thereby enabling long-horizon modeling of future world states. We also introduce an efficient and adaptive text reasoning mechanism that utilizes additional social knowledge and reasoning capabilities to further improve driving performance in challenging long-tail scenarios. We present a novel, efficient, and effective approach that achieves state-of-the-art (SOTA) results on the closed-loop Bench2drive benchmark. Codes are available at: https://github.com/hotdogcheesewhite/DeepSight.

## 1. Introduction

Recent advances in Multimodal Large Language Models (MLLMs) have opened new paradigms for end-to-end autonomous driving, leveraging their rich world knowledge acquired through pretraining in massive datasets and their aligned vision–language representation spaces (Hu et al., 2026; Wang et al., 2025; Zhang et al., 2024; Luo et al., 2025). A growing body of work has explored the integration of text reasoning mechanisms to enhance model generalization and interpretability (Hwang et al., 2025; Jiang et al., 2024). To address the pervasive spatiotemporal misalignment in text reasoning, several approaches have proposed unified architectures that jointly support generation and understanding, aiming to construct world models that capture complex spatiotemporal dynamics and thereby improve decision robustness (Zeng et al., 2025; Xiong et al., 2026).

Despite the promising potential of world modeling and visual reasoning, significant challenges remain in visual representation forms and long-horizon prediction. First, existing works adopt an autoregressive approach to predict codebook-based representations, prioritizing image texture while overlooking crucial semantic information. Second, most unified models predict only short-term future observations (e.g., 0.5 seconds), offering insufficient foresight for safe trajectory planning in complex dynamic environments. Finally, current research predominantly focuses on forward-view prediction and lacks modeling of surrounding agents around the vehicle. This deficiency in spatial awareness can easily lead to safety hazards in complex interactive scenarios (Zhao et al., 2025b).

Inspired by human driving behavior, we argue that an ideal driving world model should be equipped with human-like cognitive capabilities: precise semantic understanding, accurate spatial localization, long-horizon motion modeling, and rapid response capability. To this end, we propose DeepSight, an efficient unified world model architecture for autonomous driving. DeepSight performs parallel prediction of implicit semantic features for consecutive future frames in the BEV space, enabling long-horizon modeling of future world states. Furthermore, to handle long-tail scenarios, such as yielding to emergency vehicles, we introduce an adaptive Chain-of-Thought (CoT) mechanism. This module adaptively activates the powerful logical reasoning and social commonsense capabilities of the language model on demand, effectively balancing system efficiency and performance.

We conduct comprehensive evaluations of DeepSight on Bench2Drive, a highly challenging closed-loop simulation

---
[*]Equal contribution  [1]Tsinghua University  [2]Amap, Alibaba Group  [3]Xi'an Jiaotong University  [4]Nanyang Technological University.  Correspondence to:  Hong Wang <hong_wang@tsinghua.edu.cn>.

*Proceedings of the $43^{rd}$ International Conference on Machine Learning*, Seoul, South Korea. PMLR 306, 2026. Copyright 2026 by the author(s).

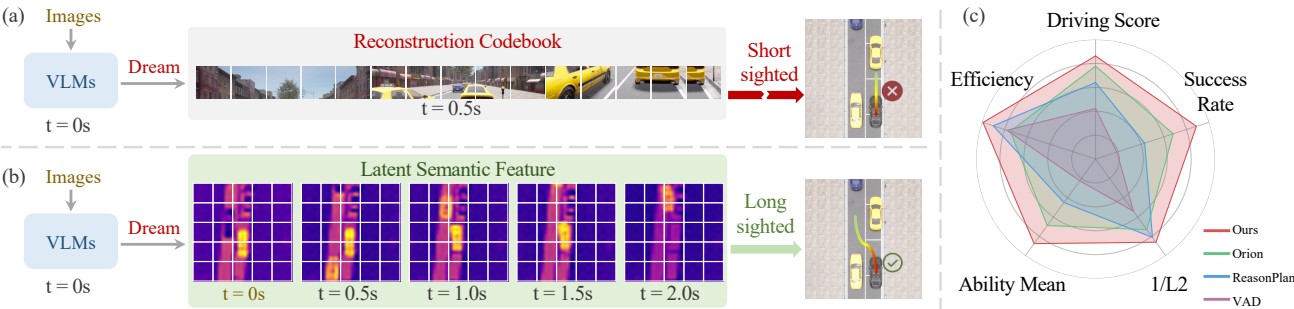

*Figure 1.* Illustration of Paradigms of Different Unified World Models. (a) the VLMs predicting images by explicitly outputting codebook tokens for future single-frame cannot support long-term prediction, this short-sightedness hinders accurate trajectory planning. (b)our VLMs achieve long-term world modeling by predicting future multi-frame latent features, enabling long-sighted planning of safe trajectories.(c)The proposed DeepSight achieves leading performance on most of metrics compared with E2E methods.

benchmark. To ensure a fair comparison, all experiments are conducted under the standard Think2Drive expert data protocol, distinguishing our results from methods that utilize different expert distributions, such as PDM-Lite. By incorporating long-horizon modeling of the future world states, our method achieves Driving Score (DS) of 84.52 (+5.68) and Success Rate (SR) of 65.91% (+8.81), substantially outperforming the latest state-of-the-art approaches. With the adaptive CoT mechanism enabled, DeepSight further improves to DS of 86.23 (+7.39) and SR of 71.36% (+13.63), with only a marginal increase in inference latency ($\sim 4\%$). Furthermore, DeepSight achieves superior performance on the open-loop nuScenes dataset.

In summary, our contributions include:

- We propose a novel driving-world model that incorporates long-horizon modeling of the future world states through parallel prediction of implicit semantic features for consecutive future frames.

- We design an adaptive CoT mechanism that effectively integrates social common sense and logical reasoning to improve driving performance in long-tail scenarios.

- DeepSight achieves SOTA performance on the Bench2Drive, validating the superiority of our approach (Jia et al., 2024).

## 2. Related Work

### 2.1. End-to-End Autonomous Driving

End-to-end autonomous driving systems (Jia et al., 2023b; Li et al., 2024b; Sima et al., 2025; Liao et al., 2024; Li et al., 2024a) directly map raw sensor inputs to driving trajectories through unified architectures, eliminating the need for handcrafted intermediate modules. Prominent examples include UniAD (Hu et al., 2023) and VAD (Jiang et al., 2023), which unify perception, motion prediction, and trajectory planning

within a single framework. HiP-AD (Tang et al., 2025) enables comprehensive interaction by allowing planning queries to iteratively interact with perception queries in the BEV space while dynamically extracting image features from perspective views. Meanwhile, generative approaches such as GenAD (Zheng et al., 2024) and GoalFlow (Xing et al., 2025b) utilize diffusion models to produce diverse, multimodal future trajectories conditioned on the context of the scene. However, these imitation learning-based systems struggle with interpretability and generalization in long-tail closed-loop scenarios (Tian et al., 2025; Renz et al., 2025). In this work, we develop a system that excels in closed-loop evaluations, generating accurate trajectories in complex driving scenarios.

### 2.2. MLLM for Autonomous Driving

Recent studies have increasingly integrated VLMs or LLMs into autonomous driving systems, motivated by the remarkable capabilities of LLMs in world knowledge, reasoning, and interpretability (Kim et al., 2024). Notably, EMMA (Hwang et al., 2024; Xing et al., 2025a) leverages the multimodal foundation of Gemini by encoding all non-sensor inputs and outputs into natural language text, fully exploiting the pre-trained LLM's world knowledge and enhancing planning through CoT textual generation. Sim-Lingo (Renz et al., 2025) incorporates a CoT Annotation mechanism into its architecture, which injects textual reasoning prior to action generation. This provides an explanatory intermediate text to mitigate hallucinations commonly observed in large models. Orion (Fu et al., 2025) bridges the gap between the reasoning space and the planning space by integrating Visual Question Answering (VQA) tasks into trajectory planning, using a dedicated generator to produce trajectories conditioned on semantic reasoning. To fully harness the reasoning capabilities of MLLMs while maintaining optimal system efficiency, We design an adaptive CoT that dynamically injects external knowledge into the reasoning process, boosting DeepSight performance in long-

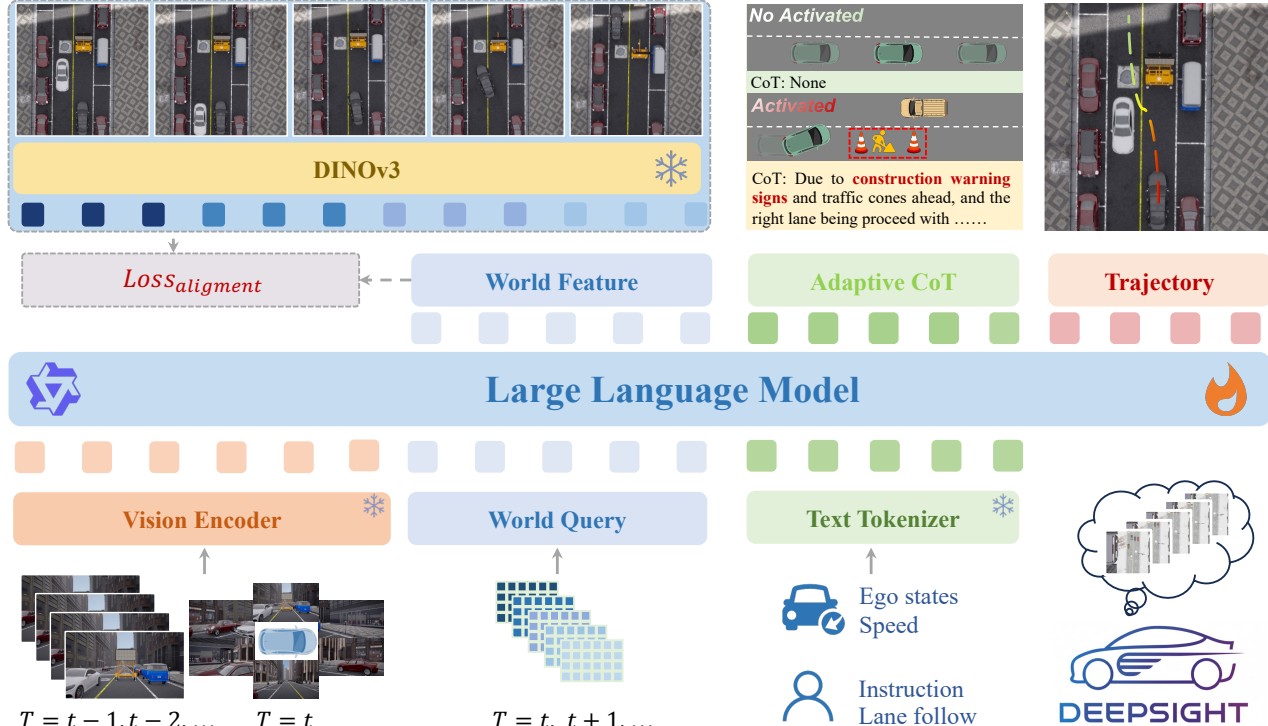

*Figure 2.* The pipeline of our method, a holistic training and inference framework for closed-loop driving. It consists of two main modules: (a) Long-term driving-world model, for aligning DINOv3 features extracted from future multi-frame RGB images in the BEV space during training. (b) An adaptive CoT module for integrating external knowledge to enhance reasoning and decision-making in long-tail cases.

tail driving scenarios.

### 2.3. World Models for Autonomous Driving

The commonly acceptable description of world models is understanding the current state and predicting the future state (Li et al., 2025a; Zhao et al., 2025a; Li et al., 2025b; Li & Cui, 2025). With the advancement of VLMs, assigning world-modeling responsibilities to VLMs has emerged as a promising research direction. This powerful architecture enables facilitates a comprehensive understanding and accurate prediction of the ego-vehicle's state, which supports decision-making and trajectory planning grounded in predicted world dynamics. FSDrive (Zeng et al., 2025) employs VLMs as its world model to generate future frames for predicting environmental dynamics and trajectory planning. ReasonPlan (Liu et al., 2025) generates the next-frame image in a self-supervised manner and combines it with textual CoT reasoning for decision-making. HERMES (Zhou et al., 2025a) employs a shared LLM to jointly drive future LiDAR point cloud generation and scene understanding. However, these world models emphasize the texture of future images, while overlooking the semantic information critical to driving tasks. Moreover, they are limited to short-term predictions, failing to enable safe and reliable trajectory planning.

This paper proposes an implicit world modeling approach built upon the VLM architecture, enabling long-horizon image feature prediction and enhancing future trajectory planning capabilities.

## 3. Proposed Method: DeepSight

The proposed DeepSight is illustrated in Figure 2. Section 3.1 describes the preliminaries. Section 3.2 elaborates on the driving-world model. Section 3.3 presents the Adaptive CoT method. Section 3.4 describes the unified training strategy and Section 3.5 details the inference pipeline.

### 3.1. Preliminary

We propose DeepSight, a unified generative-understanding framework, denoted as $M_{uni}$. Figure 2 illustrates the overall pipeline of our proposed method. DeepSight simultaneously outputs future latent features $\mathbf{F} = [f_0, f_1, f_2, f_3, f_4]$, where each $f_k \in \mathbb{R}^{h_{bev} \times w_{bev} \times d_{bev}}$ corresponds to the latent representation at time $t + k \cdot \Delta t$ ($k = 0, 1, 2, 3, 4$), CoT text $T_{cot}$, and future trajectory waypoints $\mathbf{P}_t = \{p_1, p_2, \ldots, p_n\}$. Each waypoint $p_i = (x_i, y_i)$ represents the ego-vehicle's position at time $t + i \cdot \Delta t$. Typically, the time step $\Delta t = 0.5$ s, so this sequence covers the dynamic scene evolution over the

next 2 seconds from the current moment. The trajectory waypoints $\mathbf{P}_t$ are subsequently converted into ego-vehicle control commands, such as longitudinal acceleration and steering angle.

Given $N$ multi-view image inputs at current time $t$, $\mathbf{I}_t = \{I_t^{\text{front}}, I_t^{\text{front\_left}}, \ldots, I_t^{\text{back}}\}$ where $I_t \in \mathbb{R}^{H \times W \times 3}$, along with historical frames $\mathbf{I}_{t-\tau}$, where $\tau = 1, 2, 3, 4$, the ego-vehicle state $T_{\text{ego}}$, the target point $T_{\text{target}}$, and a set of learnable World Queries $\mathbf{Q}_{\text{world}} = [q_0, q_1, q_2, q_3, q_4]$, where $q_k \in \mathbb{R}^{h_{\text{bev}} \times w_{\text{bev}} \times d_{\text{bev}}}$ corresponds to query at time $t + k \cdot \Delta t$, the model $M_{\text{uni}}$ jointly generates future latent features $\mathbf{F}$, adaptive CoT text $T_{\text{cot}}$, and the trajectory $\mathbf{P}_t$. This process is formulated as:

$$\mathbf{F}, T_{\text{cot}}, \mathbf{P}_t = M_{\text{uni}}(\mathbf{I}_t, \mathbf{I}_{t-\tau}, T_{\text{target}}, T_{\text{ego}}, \mathbf{Q}_{\text{world}}) \quad (1)$$

### 3.2. Driving-World Model

To endow the VLM with long-horizon spatial perception and future-state anticipation, we train the model to predict latent semantic features $\mathbf{F} = [f_0, f_1, f_2, f_3, f_4]$ for consecutive future frames in the BEV space. Specifically, the model extracts entity motion dynamics by processing historical frames $\mathbf{I}_{t-\tau}$ and ego-motion states $T_{\text{ego}}$, while concurrent multi-view images $\mathbf{I}_t$ provide the basis for spatial modeling. To achieve human-like efficiency in predictive modeling, we introduce $\mathbf{Q}_{\text{world}}$ as a set of learnable queries to the model. This enables the parallel inference of motion states across multiple future time steps within a single forward pass. This modeling of long-term temporal dynamics in the BEV perspective is expressed as:

$$\mathbf{F} = M_{\text{uni}}(\mathbf{I}_t, \mathbf{I}_{t-\tau}, T_{\text{target}}, T_{\text{ego}}, \mathbf{Q}_{\text{world}}) \quad (2)$$

**Ground Truth Construction:** To ensure the acquisition of robust semantic representations, we employ DINOv3 $\phi_{\text{dino}}$ as a semantic feature extractor to construct the ground truth $\mathbf{F}'$. DINOv3's potent self-supervised representation capabilities allow it to capture nuanced semantic information. The target features are defined as:

$$f_i = \phi_{\text{dino}}(\mathbf{I}_i^{bev}) \quad (3)$$

where $\mathbf{I}_i^{bev}$ represents either BEV-rendered images or semantic segmentation maps, both satisfy the model's requirements, as the model focuses on accurately characterizing the semantic and spatial distribution of the environment.

### 3.3. Adaptive CoT Method

When encountering intricate traffic participants or long-tail scenarios (e.g., complex traffic light logic, directional signage analysis, Out-of-Distribution construction zones, or emergency vehicle yielding), the model may need to invoke external knowledge. To address this, we introduce an **Adaptive Chain-of-Thought** mechanism: after observing all inputs and modeling the future world state $\mathbf{F}$, the model autonomously determines whether to activate CoT for enhanced reasoning. The adaptive CoT generation is formulated as:

$$T_{\text{cot}} = M_{\text{uni}}(\mathbf{I}_t, \mathbf{I}_{t'}, T_{\text{target}}, T_{\text{ego}}, \mathbf{Q}_{\text{world}} \mid \mathbf{F}) \quad (4)$$

If the model activates CoT, it generates a structured thought text; otherwise, it outputs a designated placeholder token $T_{\text{cot}}^{\emptyset}$ to minimize computational overhead.

**Ground Truth Annotation:** Due to the scarcity of standardized adaptive CoT datasets for Bench2Drive, we developed an automated high-fidelity labeling pipeline based on Qwen3-VL-235B. This pipeline comprises three core components: (1) scene complexity assessment, (2) complexity-based external knowledge retrieval, and (3) driving behavior determination. Using this pipeline, we synthesized approximately 1.3M high-precision structured annotations for the Bench2Drive dataset, which will be open-sourced.

### 3.4. Unified Training Strategy

To ensure the model better understands trajectory generation, we encode trajectory waypoints into *action tokens* based on their pixel-space coordinates in the BEV grid. Specifically, each waypoint $p_i = (x_i, y_i)$ is quantized and mapped to a discrete token index $t_i \in \{1, 2, \ldots, K\}$, where $K$ is the total number of grid cells. This encoding allows us to compute a Cross-Entropy (CE) loss between predicted and ground-truth tokens, as both are represented in the same tokenized space.

The model, adopting a unified generative-understanding framework, is trained end-to-end by minimizing a composite loss function $L$, which is a weighted sum of three components: trajectory loss $L_{\text{traj}}$, CoT loss $L_{\text{cot}}$, and world state prediction loss $L_{\text{world}}$:

$$L = \lambda_{\text{traj}} L_{\text{traj}} + \lambda_{\text{cot}} L_{\text{cot}} + \lambda_{\text{world}} L_{\text{world}} \quad (5)$$

where:

- $L_{\text{traj}} = \text{CE}(\mathbf{T}_{\text{traj}}, \mathbf{T}_{\text{traj}}^{\text{gt}})$ denotes the cross-entropy loss over predicted and ground-truth trajectory tokens (as described above),

- $L_{\text{cot}} = \text{CE}(\mathbf{T}_{\text{cot}}, \mathbf{T}_{\text{cot}}^{\text{gt}})$ represents the cross-entropy loss for token-level CoT text generation,

- $L_{\text{world}} = \text{MSE}(\mathbf{F}, \mathbf{F}^{\text{gt}})$ signifies the mean squared error loss between predicted latent features $\mathbf{F}$ and DINOv3-extracted ground-truth world state features $\mathbf{F}^{\text{gt}}$.

*Table 1.* Closed-loop Results of E2E-AD Methods in Bench2Drive under base set. * denote expert feature distillation. DS: Driving Score, SR: Success Rate. Red values denote the performance gains over the latest SOTA method under the same Think2Drive protocol. Methods in gray utilize different expert distributions (PDM-Lite) and are provided for reference only.

| Method | Paradigm | Expert | Closed-loop | | | |
|---|---|---|---|---|---|---|
| | | | DS↑ | SR(%)↑ | Efficiency↑ | Comfortness↑ |
| TCP*(Wu et al., 2022) | E2E | Think2Drive | 40.70 | 15.00 | 54.26 | 47.80 |
| TCP-ctrl* | E2E | Think2Drive | 30.47 | 7.27 | 55.97 | 51.51 |
| TCP-traj* | E2E | Think2Drive | 59.90 | 30.00 | 76.54 | 18.08 |
| TCP-traj w/o distillation | E2E | Think2Drive | 49.30 | 20.45 | 78.78 | 22.96 |
| ThinkTwice*(Jia et al., 2023b) | E2E | Think2Drive | 62.44 | 31.23 | 69.33 | 16.22 |
| DriveAdapter*(Jia et al., 2023a) | E2E | Think2Drive | 64.22 | 33.08 | 70.22 | 16.01 |
| VAD(Jiang et al., 2023) | E2E | Think2Drive | 42.35 | 15.00 | 157.94 | 46.01 |
| GenAD(Zheng et al., 2024) | E2E | Think2Drive | 44.81 | 15.90 | - | - |
| MomAD(Song et al., 2025) | E2E | Think2Drive | 44.54 | 16.71 | 170.21 | 48.63 |
| DriveTrans(Jia et al., 2025) | E2E | Think2Drive | 63.46 | 35.01 | 100.64 | 20.78 |
| ReasonPlan(Liu et al., 2025) | VLM | Think2Drive | 64.01 | 34.55 | 180.64 | 25.63 |
| ORION(Fu et al., 2025) | VLM | Think2Drive | 77.74 | 54.62 | 151.48 | 17.38 |
| AutoVLA(Zhou et al., 2025b) | VLM | Think2Drive | 78.84 | 57.73 | 146.93 | 39.33 |
| DiffusionDrive(Liao et al., 2024) | E2E | PDM-Lite | 77.68 | 52.72 | - | - |
| SimLingo(Renz et al., 2025) | VLM | PDM-Lite | 85.94 | 66.82 | 244.18 | 25.49 |
| DeepSight w/o adaptive CoT (**Ours**) | VLM | Think2Drive | **84.52** (+5.68) | **65.91** (+8.81) | **198.80** (+18.16) | 14.25 |
| DeepSight (**Ours**) | VLM | Think2Drive | **86.23** (+7.39) | **71.36** (+13.63) | **201.71** (+21.07) | 16.11 |

Here, $\lambda_{\text{traj}}$, $\lambda_{\text{cot}}$, and $\lambda_{\text{world}}$ are hyperparameters that balance the relative importance of trajectory prediction, CoT reasoning, and world state modeling, respectively.

### 3.5. Inference Pipeline

During inference, the input sequence $\mathcal{X}$, comprising historical images $\mathbf{I}_{t-\tau}$, surround-view images $\mathbf{I}_t$, ego-vehicle states $T_{\text{ego}}$, target waypoints $T_{\text{target}}$, and learnable queries $\mathbf{Q}_{\text{world}}$, are pre-filled as inputs to the model, where they interact and fuse through deep self-attention mechanisms. The model first performs parallel predict long-horizon future features $p(\mathbf{F} \mid \mathcal{X})$ for the coming seconds, then adaptively activates the CoT process $p(T_{\text{cot}} \mid \mathcal{X}$ based on scene demand, and finally output the future trajectory $p(\mathbf{P}_t \mid \mathcal{X}, \mathbf{F}, T_{\text{cot}})$. Notably, these three outputs are generated within a single unified forward pass, which can be formally expressed as:

$$p(\mathbf{P}_t, T_{\text{cot}}, \mathbf{F} \mid \mathcal{X}) =$$
$$p(\mathbf{F} \mid \mathcal{X}) \cdot p(T_{\text{cot}} \mid \mathcal{X}, \mathbf{F}) \cdot p(\mathbf{P}_t \mid \mathcal{X}, \mathbf{F}, T_{\text{cot}}) \quad (6)$$

Owing to our unified generative-understanding architecture, the entire inference pipeline requires no additional external generative models. Moreover, the design of parallel latent feature decoding combined with adaptive CoT ensures that our model incurs negligible additional time overhead compared to a native VLM architecture.

## 4. Experiments

### 4.1. Experiment Settings

**Datasets.** We trained and evaluated DeepSight on the Bench2Drive dataset, a closed-loop evaluation protocol under CARLA V2 for E2E autonomous driving. Bench2Drive provides 10,000 sampled driving segments for training and offers closed-loop evaluation environments with detailed simulator configurations (Jia et al., 2024). Each segment captures approximately 150 meters of continuous driving within a specific traffic scenario. We evaluated the proposed method on Bench2Drive's official set of 220 short routes, which span 44 distinct interactive scenarios, with 5 routes per scenario.

**Metrics.** Bench2drive includes five metrics for closed-loop evaluation: Driving Score (DS), Success Rate(SR), Efficiency, Comfortness, and Multi-Ability. The Success Rate quantifies the proportion of routes successfully completed within the allotted time. The Driving Score follows CARLA, incorporating both route completion status and violation penalties, where infractions reduce the score via discount factors (Dosovitskiy et al., 2017). Efficiency and Comfortness are used to measure the speed performance and comfort of the autonomous driving system during the driving process, respectively. Multi-Ability measures 5 advanced skills independently for urban driving. For ablation studies, we focus on three key metrics: Route Completion (RC), Infraction Score (IS), and Driving Score (DS), to analyze the contribution of different world modeling methods to overall driving performance.

**Implementation.** We employ Qwen2.5-VL-3B (Bai et al., 2025) as our base VLM. All experiments are conducted on 64 NVIDIA H20 GPUs with 96 GB of memory. During training, we use $2 \times 10^{-5}$ learning rate and batch size 128, for 2 epochs in Bench2Drive. Additional detailed information is listed in the *Appendix*.

## 4.2. Main Results

As shown in Table 1, DeepSight achieves the best closed-loop performance, significantly outperforming all existing E2E and VLM methods on Bench2Drive. Our approach substantially improves both SR and DS. In open-loop evaluation, our L2 error is reduced to 0.58, demonstrating superior future trajectory prediction accuracy. Specifically, DeepSight surpasses the latest SOTA method AutoVLA(Zhou et al., 2025b) by +7.39 DS and +13.63% SR. DeepSight also achieves the highest efficiency score of 201.71 among all evaluated methods, demonstrating an effective and proactive driving policy. The comfort score of 16.11 reflects a common trade-off between trajectory agility and smoothness. Nonetheless, the comfort margin remains within acceptable bounds and could be further optimized via post-smoothing or low-level controller tuning. Moreover, even without incorporating textual reasoning, our method, relying merely on the world modeling, outperforms the latest SOTA method.

Additionally, multi-ability results are reported in Table 2. Our model excels in scenarios such as overtaking (91.11%) and emergency braking (78.33%), indicating that in tasks predominantly requiring intuitive reasoning, our strong long-horizon BEV spatial modeling enables smooth and robust handling. Moreover, in challenging multi-vehicle interaction scenarios, our model shows significant improvements over conventional methods, demonstrating its powerful capability to model causal relationships among the ego vehicle, dynamic agents, and static elements in complex driving environments.

## 4.3. Analysis of Different World Modeling

**Explicit Reconstruction vs. Latent Semantic**   Table 3 presents the closed-loop evaluation results of different world modeling approaches. We believe that an ideal world model should possess precise semantic understanding, accurate spatial localization, and long-horizon motion modeling. Therefore, we first compare VAE-based world modeling approaches (Zeng et al., 2025) that use codebooks to focus on texture features with latent-feature-based methods such as DeepSight (ID 1 and ID 2). The results demonstrate that our world modeling approach, exemplified by DeepSight, achieves a significant improvement in DS (+47.04), validate that our world modeling method effectively captures and interprets the semantic structure of driving environments.

**Short-horizon vs. Long-horizon**   Furthermore, our analysis of different temporal modeling settings (ID 1 and ID 3) reveals that when the VAE is used to encode five consecutive frames before predicting a trajectory, the driving score drops significantly ($-13.09$ DS). This indicates that the VAE is fundamentally inadequate for modeling long-horizon world relationships. In contrast, our latent world modeling approach (ID 2 and ID 4) already demonstrates strong performance under the single-frame trajectory prediction paradigm. Moreover, when extended to multi-frame prediction, it further boosts the driving score by (+11.78 DS). This clearly demonstrates that our model has a significant advantage in capturing long-horizon world dynamics, enabling it to effectively model temporal relationships spanning extended time periods.

**Forward-View vs. Bird's-Eye-View**   As we mentioned, an ideal world model should be capable of precisely perceiving spatial structure. To validate the advantage of BEV spatial prediction, we evaluated our model under different observational perspectives, specifically, by applying the same processing pipeline to front-view inputs. As shown in Table 4, the results show that even in the front-view setting, our paradigm remains effective, achieving strong performance on the 10-route benchmark (DS: 77.77, IS: 0.87). However, the front-view perspective inherently lacks comprehensive modeling of surrounding vehicles and agents, which constrains the model's ability to perform safe, long-horizon planning. In contrast, when we adopt BEV-based spatial modeling, the driving score improves significantly (+8.8 DS). This outcome not only confirms the effectiveness of BEV-centric spatial reasoning but also demonstrates the strong generalizability of our proposed paradigm across diverse observational viewpoints.

## 4.4. Analysis of Adaptive COT

**Effectiveness of Adaptive COT**   Table 5 provides a detailed ablation study of the adaptive CoT mechanism. By comparing ID 1, ID 2, and ID 3, we show that while the adaptive CoT does improve SR, this gain is less significant than the improvement achieved through driving world model (ID 3), revealing the inherent limitations of adaptive CoT alone in autonomous driving scenarios. Compared to ID 3, ID 4 achieves the best overall performance, confirming the complementary nature of driving world model and adaptive textual reasoning.

**Qualitative Visualization**   We validate the effectiveness of Adaptive CoT in long-tail driving scenarios through three canonical closed-loop evaluation settings. Figure 3 presents our model's outputs for driving behavior reasoning and trajectory prediction, along with corresponding ego-vehicle states. We observe that, in these three complex scenarios,

*Table 2.* Multi-Ability Results of E2E-AD Methods under base set. * denote expert feature distillation.

| Method | Paradigm | Merging | Overtaking | Emergency Brake | Give Way | Traffic Sign | Mean |
|---|---|---|---|---|---|---|---|
| | | | | Ability (%) ↑ | | | |
| TCP* (Wu et al., 2022) | E2E | 16.18 | 20.00 | 20.00 | 10.00 | 6.99 | 14.63 |
| TCP-ctrl* | E2E | 10.29 | 4.44 | 10.00 | 10.00 | 6.45 | 8.23 |
| TCP-traj* | E2E | 8.89 | 24.29 | 51.67 | 40.00 | 46.28 | 34.22 |
| TCP-traj w/o distillation | E2E | 17.14 | 6.67 | 40.00 | **50.00** | 28.72 | 28.51 |
| ThinkTwice* (Jia et al., 2023b) | E2E | 27.38 | 18.42 | 35.82 | **50.00** | 54.23 | 37.17 |
| DriveAdapter* (Jia et al., 2023a) | E2E | 28.82 | 26.38 | 48.76 | **50.00** | 56.43 | 42.08 |
| AD-MLP (Zhai et al., 2023) | E2E | 0.00 | 0.00 | 0.00 | 0.00 | 4.35 | 0.87 |
| UniAD-Tiny (Hu et al., 2023) | E2E | 8.89 | 9.33 | 20.00 | 20.00 | 15.43 | 14.73 |
| UniAD-Base (Hu et al., 2023) | E2E | 14.10 | 17.78 | 21.67 | 10.00 | 14.21 | 15.55 |
| VAD (Jiang et al., 2023) | E2E | 8.11 | 24.44 | 18.64 | 20.00 | 19.15 | 18.07 |
| ReasonPlan (Liu et al., 2025) | VLM | 37.50 | 26.67 | 33.30 | 40.00 | 45.76 | 36.66 |
| DriveTrans (Jia et al., 2025) | VLM | 17.57 | 35.00 | 48.36 | 40.00 | 52.10 | 38.60 |
| ORION (Fu et al., 2025) | VLM | 25.00 | 71.11 | 78.33 | 30.00 | 69.15 | 54.72 |
| DeepSight (**Ours**) | VLM | **60.00** | **91.11** | **78.33** | **50.00** | 71.58 | **70.20** (+15.48) |

*Table 3.* Closed-loop evaluation results in Dev 10 routes for different world modeling methods.

| ID | TYPE | FRAME | RC↑ | IS↑ | DS↑ |
|---|---|---|---|---|---|
| 1 | VAE | ONE | 47.56 | 0.64 | 27.75 |
| 2 | DINOV3 | ONE | 90.49 | 0.83 | 74.79 |
| 3 | VAE | FIVE | 27.02 | 0.66 | 14.66 |
| 4 | DINOV3 | FIVE | **95.95** | **0.89** | **86.57** |

*Table 4.* Closed-loop evaluation results in Dev 10 routes for different view alignment.

| ID | TYPE | RC↑ | IS↑ | DS↑ |
|---|---|---|---|---|
| 1 | FRONT VIEW | 89.47 | 0.87 | 77.77 |
| 2 | BEV VIEW | **95.95** | **0.89** | **86.57** |

*Table 5.* Ablation results of CoT type in Bench2drive 220 routes. DS and SR denote Driving Score and Success Rate.

| ID | WM | ADA-COT | DS↑ | SR↑ | Eff↑ |
|---|---|---|---|---|---|
| 1 | ✗ | ✗ | 58.16 | 28.18 | 190.76 |
| 2 | ✗ | ✓ | 69.87 | 42.27 | 187.39 |
| 3 | ✓ | ✗ | 84.52 | 65.91 | 198.80 |
| 4 | ✓ | ✓ | **86.23** | **71.36** | **201.71** |

*Table 6.* Comparison of inference speeds among different end-to-end methods

| METHOD | WM | COT | ADD. TIME%↓ |
|---|---|---|---|
| QWEN2.5-VL | ✗ | ✗ | 0 |
| FSDRIVE | ✓ | ✗ | +60.71 |
| DEEPSIGHT | ✓ | ✗ | +3.57 |
| DEEPSIGHT | ✓ | ✓ | +7.69 (+4.12) |

DeepSight dynamically assesses scene complexity and determines whether to activate social knowledge reasoning, the resulting text reasoning outputs are shown on the right. DeepSight effectively recognizes critical environmental elements such as STOP signs, fire trucks, and traffic cones that are essential for rational decision-making. Then it infers appropriate responses, including deceleration or other safety-oriented maneuvers. This demonstrates that incorporating contextually relevant social knowledge significantly enhances the robustness and safety of autonomous driving systems in highly dynamic and complex real-world environments. Additional qualitative results and visual examples are provided in the *Appendix*.

### 4.5. Analysis of Additional Time Overhead

We also conducted ablation studies to evaluate the additional inference latency introduced by the world model and the adaptive CoT mechanism. We selected a fine-tuned Qwen2.5-VL model as our baseline, which takes the same

inputs and directly outputs trajectories. As illustrated in Table 6, we compare the inference overhead of different world model designs. DeepSight introduces only a 3.57% additional latency compared to the native VLM, validating the efficiency of our parallel latent future state prediction paradigm. Moreover, thanks to the conditional activation of the CoT module, triggered only when necessary, the average inference latency increases by only 4.12%. These results demonstrate the high efficiency of both the driving-world model and the adaptive CoT within DeepSight.

## 5. Conclusion

This paper presents DeepSight, an autonomous driving framework based on latent feature prediction, enabling VLMs to effectively model the world. Through world modeling, our DeepSight captures long-term world states in dynamic driving scenarios. In addition, by introducing adaptive CoT, DeepSight efficiently integrates social knowledge

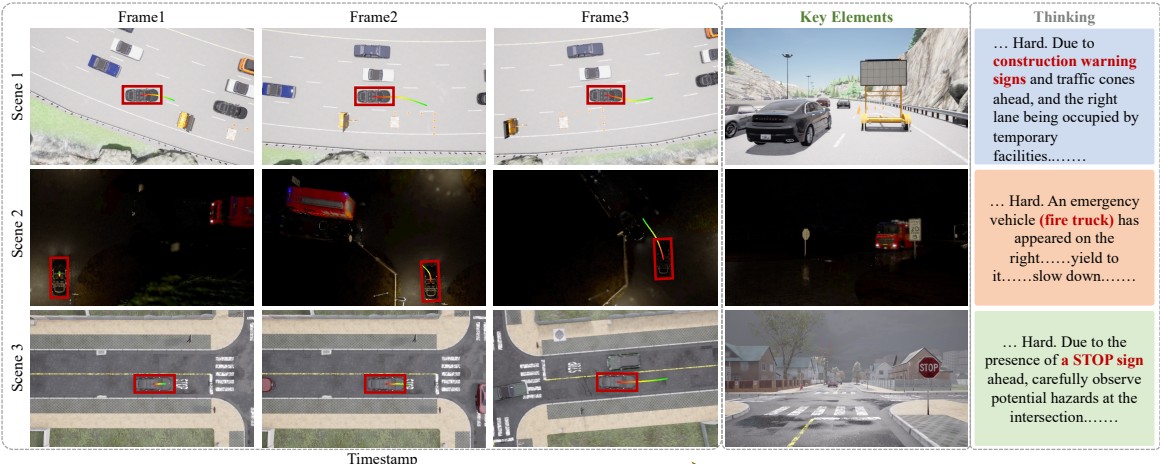

*Figure 3.* Qualitative results of DeepSight on the Bench2Drive closed-loop evaluation set. The three rows of the figure depict three distinct driving scenarios. The first three columns present temporally consecutive BEV frames, with red bounding boxes indicating the ego vehicle's position. The fourth column displays corresponding PV images that highlight critical frame for safe driving. The final column presents the model-generated CoT output, which analyzes scene-specific information, including construction zones, emergency vehicles, and traffic signs, thereby effectively enhancing the safety of trajectory planning.

into driving decisions in long-tail scenarios. Extensive experimental results demonstrate the effectiveness of the proposed DeepSight approach, advancing autonomous driving toward world modeling.

**Limitations and Future Work** Although DeepSight performs well in the Bench2Drive closed-loop simulation environment (Jia et al., 2024), it is constrained by the high computational complexity of scaling VLMs to real-time driving scenarios. In the future, We plan to explore more efficient driving world models, further enhancing the model's generalization capability and reasoning speed.

## Impact Statement

This paper presents work whose goal is to advance the field of Machine Learning. There are many potential societal consequences of our work, none which we feel must be specifically highlighted here.

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

## A. Reasoning Annotation Pipeline

To provide high-quality, adaptive reasoning data for the SFT phase, we developed an automated data construction pipeline consisting of a two-stage workflow. Given inputs with the current camera, history video, driving command, and instruction, the powerful MLLM Qwen3-VL-235B first generates a raw text CoT. Figure 4 illustrates a complete sample of the annotation dataset, detailing the reasoning steps. The prompt for Qwen3-VL-235B is shown in Figure 5. The text CoT then passes through two-stage. This pipeline systematically processes raw datasets that contain only multimodal inputs and converts them into comprehensive reasoning datasets, effectively integrating the model's understanding of scene complexity and distilling Qwen3vl-235B's scene-understanding capability.

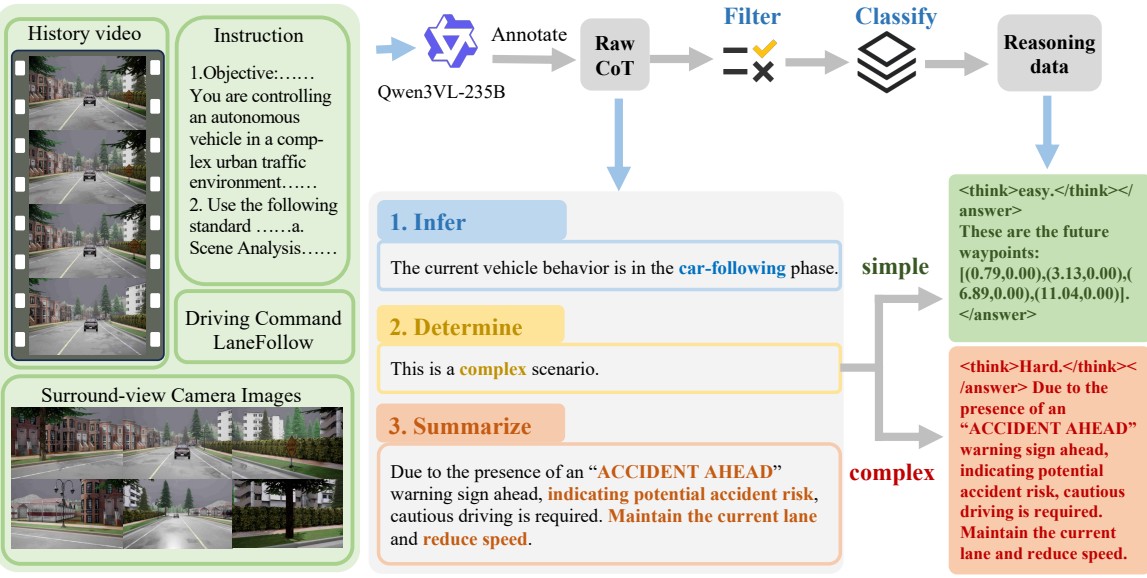

*Figure 4.* A complete sample of the annotation dataset.

- **Format Filter:** This rule-based filter checks whether the text reasoning is composed of the three parts: (1) Infer current action based on historical information, (2) Determine whether complex decision-making is required based on road conditions, and (3) Summarize inference results. Furthermore, we examined and filtered two types of errors: when the model judged a scenario to be simple but still provided some reasoning, and when it judged a scenario to be complex but did not provide any reasoning. This is crucial for our next step of assembly.

- **Classify:** In this process, we primarily categorize the annotated data. We carefully designed how the model produces its final answers. First, we require the model to make a judgment about the current vehicle's motion state, which helps it better understand whether to accelerate or decelerate when making the final decision. In our prompt, we ask the model to determine the vehicle's motion state solely from changes in surrounding pixels. We also provide multiple conditions for determining when external knowledge is needed, to assist the model in making precise judgments. However, we believe these two steps do not need to be fully distilled into our DeepSight; we only aim to retain the key reasoning part. Therefore, we classified Qwen3-VL-235B's answers: if "simple" appears during the assessment of scene complexity, we regard it as a simple scene and directly assemble the real trajectory values; if "complex" appears, we regard it as a complex scene and extract only the third part—the reasoning in the summary—to help the model provide decision information.

## B. Implementation Details

All experiments are conducted on 64 NVIDIA H20 GPUs (96 GB each). We employ Qwen2.5-VL-3B (Bai et al., 2025) as our base VLM. During SFT, we use $2 \times 10^{-4}$ learning rate and batch size of 64, for 2 epochs in Bench2Drive. The vision encoder of DeepSight is frozen, and the LLM is fully fine-tuned in the SFT stage. We observe that DINOv3-ViT-L/16 is a vision model based on the Vision Transformer (ViT) architecture within the DINOv3 series, specifically designed

1. **Objective:**
   You are controlling an autonomous vehicle in a complex urban traffic environment. Currently available inputs include: images from six camera views (front, rear, left, right, front-left, front-right), the ego vehicle's trajectory over the past 2 seconds, and historical images from the CAM_FRONT camera over the past 2 seconds. Your task is to combine driving command and plan safe and reasonable driving behavior for the ego vehicle over the next 2 seconds. Please perform reasoning and output in Chinese.

2. **Process Overview:**
   Please strictly follow the two steps below sequentially for reasoning:
   a. **[Infer current action based on historical information]**: Based on pixel features, pay attention to lane markings, reference objects, etc., and determine which of the following stages the current vehicle behavior is in: [Following, Lane changing, Turning left, Turning right, Stopping and waiting, Other]. Output requirement: Strictly select from the provided options; format: "Current vehicle behavior is in [stage]." (Example: "Current vehicle behavior is in lane changing stage."). If no match is found in the list, choose "Other".
   b. **[Determine whether complex decision-making is required based on road conditions]**: If the current environment involves multi-object interaction (e.g., intersection passage, overtaking obstacle vehicles, merging into main road, yielding to emergency vehicles, etc.), determine whether it qualifies as a "complex decision scenario." Judgment criteria:
   - If at an intersection with green light in the ego vehicle's direction and no vehicle ahead → prioritize acceleration;
   - If the preceding vehicle's tail lights flash continuously (not turn signals) without other anomalies → likely a malfunctioning vehicle, treat as complex scenario;
   - If at an uncontrolled intersection requiring turning or crossing → treat as complex scenario;
   - If a "STOP" sign is present ahead with potential risks nearby (e.g., pedestrians, cross traffic) → treat as complex scenario;
   - For other cases, apply driving common sense: if no interaction risk, classify as simple scenario; otherwise, complex.
     Output requirement : Output only "Complex scenario" or "Simple scenario". (Example: "Complex scenario")
   c. **[Summarize inference results]**: Based on the above reasoning, select a safe and reasonable future driving action from the candidate action set:
   - Direction control (choose one of five): [Maintain current lane, Change lane left, Change lane right, Turn left, Turn right]
   - Speed control (choose one of five): [Stop and wait, Decelerate, Maintain current speed, Accelerate, Overtake via adjacent lane]
     Output requirement:
   - If simple scenario: directly output action combination, format: "Direction control, Speed control" (Example: "Maintain current lane, Decelerate")
   - If complex scenario: first state the reason, then output action combination, format: "Due to [reason], [Direction control, Speed control]" (Example: "Due to STOP sign ahead and pedestrian crossing, Stop")

3. **Output Format Example:**
   [Infer current action based on historical information]:
   Current vehicle behavior is in following stage.
   [Determine whether complex decision-making is required based on road conditions]:
   Simple scenario.
   [Summarize inference results]:
   Maintain current lane, Maintain current speed

*Figure 5.* Prompt for CoT annotation by Qwen3-VL-235B

for image feature extraction and downstream vision tasks. Trained on the large-scale LVD-1689M dataset, this model demonstrates robust global modeling capabilities, making it highly suitable for dense prediction tasks such as classification and object detection. Its performance compares favorably to the much larger ViT-7B/16 model, achieving superior results across various benchmarks and significantly outperforming competing models, particularly in dense prediction tasks. As an efficient and widely adopted variant, ViT-L/16 strikes a balance between computational resources and performance, rendering it an effective choice for feature extraction in our long-term spatiotemporal modeling.

*Table 7.* Model parameters of image encoder model (Vision Transformer).

| MODEL | LAYERS | HIDDEN SIZE | NUM HEADS | PATCH SIZE |
|---|---|---|---|---|
| QWEN2.5-VL | 32 | 1280 | 16 | 14 |

## C. More Visualization

In the simulation dataset, numerous scenarios require world reasoning, such as pedestrians crossing the road, narrow roads, and other extreme conditions. DeepSight successfully handles all these challenges, as illustrated in Figure 6.

As shown in the visualizations, our extensive closed-loop testing on the simulation dataset exposes the model to highly challenging environments. The dataset includes substantial out-of-distribution data, ranging from adverse weather conditions

*Table 8.* Model parameters of LLM.

| MODEL | LAYERS | HIDDEN SIZE | KV HEADS | HEAD SIZE |
|---|---|---|---|---|
| QWEN2.5-VL | 36 | 2048 | 2 | 128 |

*Table 9.* Model parameters of DINOv3-ViT-L/16.

| MODEL | PARAMETERS | HIDDEN SIZE | PRETRAINING DATASET | PATCH SIZE |
|---|---|---|---|---|
| DINOV3VITMODEL | 300M | 1024 | LVD-1689M | 16 |

with wet-road reflections (top row) to drastic lighting variations. The model demonstrates remarkable robustness in safety-critical situations. For instance, it effectively anticipates and reacts to dynamic agents — such as pedestrians jaywalking across the street. Notably, in complex narrow-road scenarios (bottom row), the model accurately predicts the speed of oncoming vehicles and preceding traffic participants, strictly adheres to traffic rules, and makes socially compliant decisions — such as swiftly overtaking by borrowing the opposite lane when safe to do so. This behavior significantly enhances the rationality and safety of the planning logic. Although the Adaptive CoT module contributes a marginal gain to the DS, it plays a disproportionately critical role in specific long-tail scenarios. Crucially, we did not blindly stack CoT modules. Instead, recognizing the limitations of world modeling in certain contexts and carefully weighing the logical reasoning strengths of CoT against its computational latency, we devised an efficient adaptive mechanism to holistically refine our algorithmic framework. Statistical analysis of our closed-loop evaluation across 220 routes reveals that this reasoning mechanism was triggered in less than 30% of frames, ensuring efficiency.

As shown in Figure 7, we visualize DeepSight's predicted future features, revealing that essential environmental cues are successfully preserved across multiple frames. The quality of our world prediction is further validated through the visualization of DINO features. Unlike standard methods that often suffer from feature blurring over time, DeepSight generates high-fidelity future states where the road geometry remains stable and well-defined. Notably, the model tracks the motion trajectories of other vehicles and the ego-vehicle with high precision. The clear spatial localization of these agents in the feature space confirms that DeepSight not only understands the static environment but also possesses a robust understanding of agent-centric motion, providing a solid foundation for downstream autonomous driving tasks.

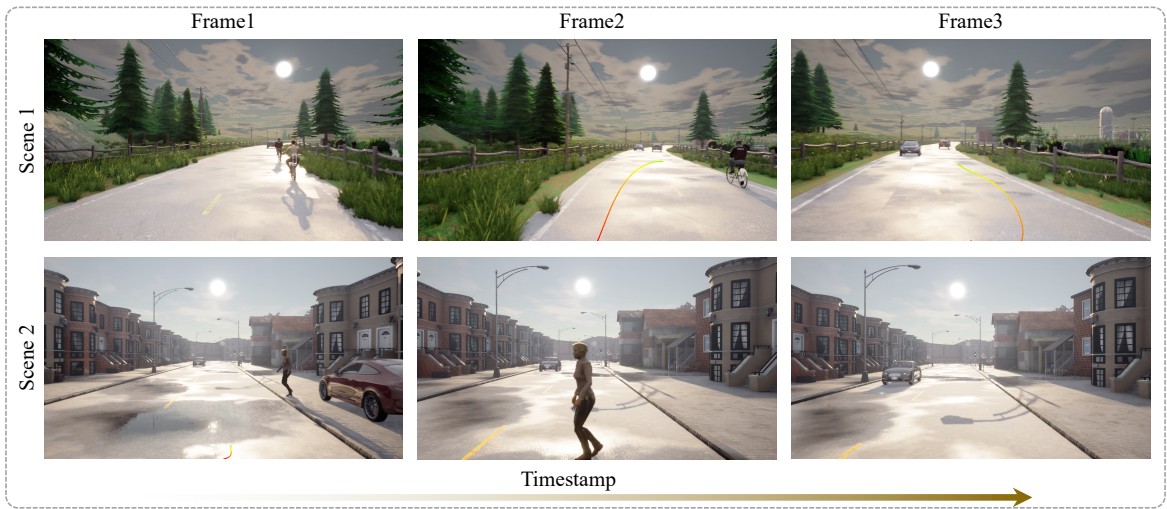

*Figure 6.* Qualitative results of DeepSight on the Bench2Drive closed-loop evaluation set.

# D. Sensitive Analysis

Since both trajectory waypoints and CoT reasoning are encoded as text tokens, they exhibit similar gradient magnitudes and convergence behaviors. We calibrate both terms using a shared hyperparameter $\lambda_{cot} = \lambda_{traj} = 1.0$. To evaluate

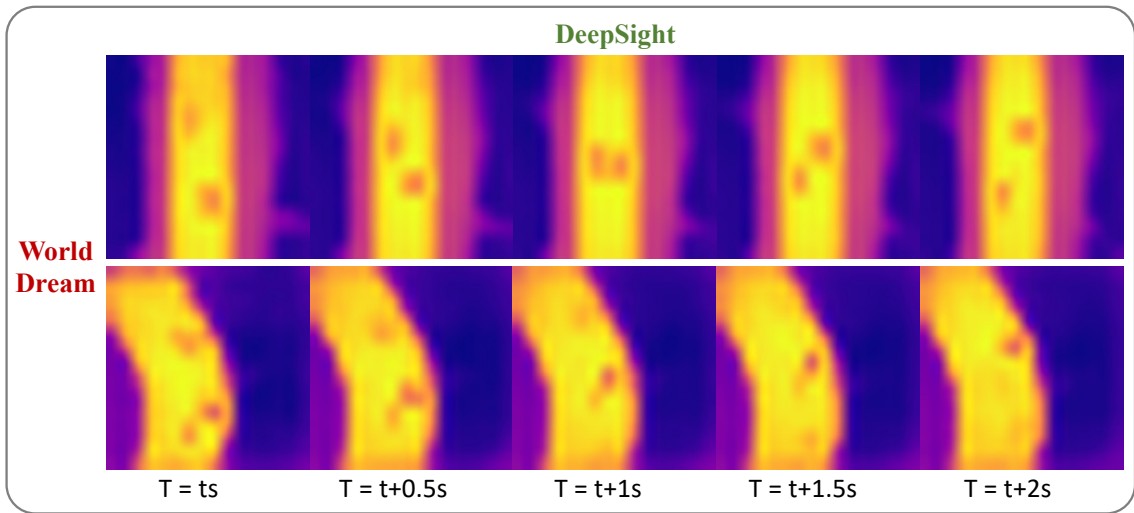

*Figure 7.* Qualitative results of DeepSight's world prediction.

hyperparameter sensitivity on Dev 10 set, as shown in Table 10, we varied $\lambda_{world} \in \{0.5, 1.0, 2.0\}$.

*Table 10.* Sensitivity analysis of the hyperparameter $\lambda_{world}$. The bold values indicate the best performance.

| $\lambda_{world}$ | RC↑ | IS↑ | DS↑ |
|---|---|---|---|
| 0.5 | 88.23 | 0.89 | 78.93 |
| **1.0** | **95.95** | **0.89** | **86.57** |
| 2.0 | 89.12 | 0.89 | 79.82 |

Interestingly, our experiments demonstrate a non-monotonic relationship where performance peaks and then drops as $\lambda$ increases. We attribute this phenomenon to the inherent ambiguity of world modeling. While an excessively high weight can degrade final performance, it still outperforms scenarios with minimal world modeling.

## E. Evaluation on Real Data

Following (Jiang et al., 2023; Zeng et al., 2025), we evaluate open-loop trajectory planning and future frames generation on the nuScenes (Caesar et al., 2020). The nuScenes contains 1,000 scenes of approximately 20 seconds each captured by six cameras providing 360-degree field of view. Specifically, the dataset provides 28,130 (train) and 6,019 (val) samples. As shown in Table 11, we have extended our evaluation to the nuScenes dataset, a widely adopted benchmark for real-world autonomous driving. We adhered to the standard evaluation protocols employed in recent works, utilizing metrics such as L2 and Collision. For a fair comparison, FSDrive was reproduced using the same base model and experimental settings as ours.

*Table 11.* Open-loop results on the nuScenes dataset. * denotes results using the same base model and experimental settings as ours. Best results are in **bold**.

| Method | L2 (m) ↓ | | | | Collision Rate (%) ↓ | | | |
|---|---|---|---|---|---|---|---|---|
| | 1s | 2s | 3s | Avg. | 1s | 2s | 3s | Avg. |
| VAD(Jiang et al., 2023) | 0.41 | 0.70 | 1.05 | 0.72 | 0.07 | 0.18 | 0.43 | 0.23 |
| LAW(Li et al., 2025a) | 0.26 | 0.57 | 1.01 | 0.61 | 0.14 | 0.21 | 0.54 | 0.30 |
| World4Drive(Zheng et al., 2025) | 0.23 | 0.47 | 0.81 | 0.50 | **0.02** | 0.12 | 0.33 | 0.16 |
| FSDrive*(Zeng et al., 2025) | 0.27 | 0.33 | 0.56 | 0.35 | 0.07 | 0.10 | 0.24 | 0.14 |
| DeepSight (**Ours**) | **0.16** | **0.31** | **0.52** | **0.33** | **0.02** | **0.07** | **0.27** | **0.12** |

