# OpenReview forum: "DeepSight: Long-Horizon World Modeling via Latent States Prediction for End-to-End Autonomous Driving"
_ICML.cc/2026/Conference — ICML 2026 regular_

### Official Review · Reviewer_GAZN · 2026-03-08

**Soundness:** 3
**Presentation:** 2
**Significance:** 3
**Originality:** 2
**Overall Recommendation:** 3
**Confidence:** 4

**Summary:**

This paper proposes DeepSight, a VLM-based framework for end-to-end autonomous driving that integrates latent world modeling and adaptive reasoning. The method predicts future latent BEV features to model long-horizon dynamics and conditionally activates Chain-of-Thought reasoning in complex scenarios. Experiments on the Bench2Drive closed-loop benchmark show improved performance over existing methods.

**Compliance With Llm Reviewing Policy:**

Affirmed.

**Final Justification:**

After reading the rebuttal, my concerns remain, so I will keep my score unchanged.

**Key Questions For Authors:**

1. The paper describes the proposed method as a driving world model that predicts latent future states. However, it is unclear how this differs fundamentally from existing VLM-based end-to-end driving models that also perform predictive modeling. Could the authors clarify where the world model property is explicitly manifested in the architecture or training objective, and how it differs from standard predictive representations in VLM-based E2E models?
2. The proposed method adopts Qwen2.5-VL as the base VLM. Have the authors experimented with other VLM architectures to verify the approach's generality?
3. The paper reports an ablation study on additional time overhead relative to the base Qwen model. However, from a practical perspective, it would also be important to compare the overall inference time with other end-to-end autonomous driving models. Could the authors report the full inference latency (e.g., FPS or per-frame inference time) and compare it with other representative E2E or VLM-based driving systems?

**Limitations:**

Yes.

**Strengths And Weaknesses:**

Strengths
1. The paper proposes DeepSight, which predicts future latent semantic features in BEV space to model long-horizon dynamics for end2end autonomous driving. This idea of latent state prediction for world modeling is relatively novel and aligns with emerging research on predictive representations for decision-making.
2. The method achieves state-of-the-art performance on the Bench2Drive closed-loop benchmark, with clear improvements in Driving Score and Success Rate compared with prior VLM-based approaches.

Weaknesses
1. The introduction claims several limitations of existing methods (e.g., short-horizon prediction and insufficient semantic modeling), but these claims are not strongly supported by analysis or evidence from prior work.
2. The paper describes world models as systems that “understand the current state and predict the future state.” Under this definition, many predictive models could qualify as world models. In practice, the proposed method appears to be a VLM-based end2end driving model that predicts latent features, and it is unclear how it fundamentally differs from common VLM-based end2end architectures.
3. Adaptive CoT mainly introduces a trigger to conditionally activate Chain-of-Thought reasoning. Compared to standard CoT-based approaches, the conceptual novelty and technical differences are limited.
4. Experiments are mainly conducted on a single closed-loop benchmark (Bench2Drive). While closed-loop evaluation is important, complementary open-loop evaluations would also be valuable for providing a more comprehensive assessment, which is currently missing in this work.

---

> ### Author Rebuttal · Authors · 2026-03-30
>
> We are grateful for the reviewer's positive evaluation and insightful suggestions. Regarding the specific issues highlighted for clarification, we provide detailed responses below, substantiated by comprehensive supplementary experiments.
> ***
> **_Q1: Could the authors clarify where the world model property is explicitly manifested in the architecture or training objective, and how it differs from standard predictive representations in VLM-based E2E models?_**
>
> A1: We thank the reviewer for this crucial question. Our approach differs from existing VLM-based E2E models in that **our model performs trajectory planning based on future predictions**, rather than treating prediction merely as an auxiliary **supervised task**.
> It is important to note that the joint prediction of world features and text decomposes the task into three components: (1) multi-frame DINO feature prediction, (2) CoT reasoning, and (3) waypoint prediction. This decomposition is illustrated in Equation 6.
>
> $$
> p(P_t, T_{cot}, F \mid X) =
> \\underset{\\text{World Prediction}}{\\underline{p(F \mid X)}} \\cdot
> \\underset{\\text{Reasoning}}{\\underline{p(T_{cot} \mid X, F)}} \\cdot
> \\underset{\\text{Planning}}{\\underline{p(P_t \mid X, F, T_{cot})}}
> $$
>
> The LLM's autoregressive prediction ensures that the output of each subsequent module is based on the attention computations derived from the preceding module.DeepSight first generates latent world features $F$, grounding subsequent CoT in anticipated future states rather than just current observations. Finally, trajectory tokens, $P_t$ are predicted based on the full context ($X, F, T_{\text{cot}}$). Furthermore, as demonstrated in Table 3, we conducted ablation studies to verify the impact of different prediction objectives and prediction horizons on overall performance. Comparative visualizations are available at: /https://anonymous.4open.science/r/test-C7FB.
> We have also organized a table for reviewer to compare DeepSight with existing E2E models.
> |**Method**|DeepSight|FSDrive|World4Drive|BEVWorld|LAW|
> |-|-|-|-|-|-|
> |**Representation of Future**|**DINOv3 Features**|Discrete Tokens|Physical Latents|BEV Latent Tokens|Visual Latents|
> |**Horizon**|**Current frame+Future 4 frames**|Next Frame|Future 3 frames|Future 6 frames|Next Frame|
> |**Input**|Camera-only|Camera-only|Camera-only+Trajectory Vocabulary|Camera+LiDAR|Camera-only|
> |**Modeling**|**Future States**+**World Knowledge**+**Waypoints**|Discrete tokens+Waypoints|Intention-aware Latents+Waypoints|BEV Features|Future Latents+Waypoints|
> ***
> **_Q2: The proposed method adopts Qwen2.5-VL as the base VLM. Have the authors experimented with other VLM architectures to verify the approach's generality?_**
>
> A2:
> First, we established a rigorous benchmark by adopting Qwen2.5-VL, the SOTA MLLM, as our baseline. The significant performance gains achieved atop this powerful foundation robustly validate the effectiveness of our proposed architectural components.
> Second, as highlighted in recent literature, such as FSDrive, the Qwen series has established itself as the central reasoning engine within autonomous driving architectures, owing to its exceptional multimodal reasoning abilities. By anchoring both our method and the baseline on this shared foundation, we ensure that **any observed performance disparities are attributable solely to our novel architectural design**, rather than variations in the underlying model's inherent capabilities. This strategy guarantees the fairness and homogeneity of our comparative analysis.
> |ID|Model|DS↑|SR↑|Eff↑|
> |-|-|-|-|-|
> |1|Qwen2.5-VL|58.16|28.18|190.76|
> |2|**DeepSight**|**86.23**|**71.36**|**201.71**|
> ***
> **_Q3: Could the authors report the full inference latency?_**
>
> A3: We evaluated the inference speed on an NVIDIA H20 GPU, comparing our approach with Orion. Orion achieves an inference speed of 0.8 Hz. In contrast, our DeepSight model reaches **1.1 Hz without vLLM acceleration**. Unlike autoregressive world models that generate future frames step-by-step, our model predicts all potential future states in a single forward pass. This significantly reduces the decoding time required for long-horizon modeling.
> ***
> **_Q4: Open-loop evaluations would also be valuable._**
>
> A4: Thank you for this suggestion. We have conducted evaluations on representative open-loop datasets. On the nuScenes, DeepSight achieves **superior performance compared to FSDrive and LAW**.
>
> |Method|L2 (1s) ↓|L2 (2s) ↓|L2 (3s) ↓|L2 (Avg) ↓|Col (1s) ↓|Col (2s) ↓|Col (3s) ↓|Col (Avg)↓|
> |-|:-:|:-:|:-:|:-:|:-:|:-:|:-:|:-:|
> |LAW|0.26|0.57|1.01|0.61|0.14|0.21|0.54|0.30|
> |World4Drive|0.23|0.47|0.81|0.50|0.02|0.12|0.33|0.16|
> |FSDrive*|0.27|0.33|0.56|0.35|0.07|0.10|0.24|0.14|
> |**DeepSight**|**0.16**|**0.31**|**0.52**|**0.33**|**0.02**|**0.07**|0.27|**0.12**|
> ***
> **_Q5: The CoT's novelty is limited._**
>
> A5: We acknowledge that previous work has provided us with inspiration. Please refer to our response to **Reviewer ZQnF, Q3** due to space limitations.

---

> > ### Author Rebuttal · Reviewer_GAZN · 2026-04-02
> >
> > Thanks for your clarification. However, I am still trying to better understand the distinction in Q1. My understanding is that prediction itself inherently involves modeling future states, which seems to be a common aspect of many existing methods. Could you kindly clarify what fundamentally differentiates your approach (world model) from prior VLM-based E2E models in this regard?

---

> > > ### Author Response · Authors · 2026-04-03
> > >
> > > We sincerely thank the reviewer for this insightful follow-up, which provides us with a valuable opportunity to clarify the fundamental distinctions between our approach and existing VLM based e2e methods.
> > >
> > > Our definition of a World Model in this paper, "The commonly acceptable description of world models is understanding the current state and predicting the future state", is drawn from **established academic consensus** rather than being a novel proposition.
> > >
> > > As articulated in **V-JEPA 2**:
> > > > *"To summarize, we show that joint-embedding predictive architectures learning from videos can be used to build a world model that enables understanding the physical world, predicting future states, and effectively planning in new situations"*
> > >
> > > Similarly, in **FSDrive**:
> > > > *"World models aim to infer ego status and dynamic environments from past observations to enable accurate future prediction and planning."*
> > >
> > > Guided by these definitions, the core objective of a world model is to learn underlying **physical laws**, rather than merely **predicting action outcomes**. Therefore, the fundamental distinction between the world models discussed in our work (including FSDrive, DreamZero, and DeepSight) and traditional VLM-based E2E methods lies in the **capacity for supervised learning of future physical world dynamics**.
> > >
> > > Even when generating CoT text or trajectory waypoints, existing methods essentially function as **Discriminative Policy Models**. They learn a mapping $P(P_t, T_{cot} \mid X)$, representing a discriminative decision process that predicts **the probability of the most likely token** under specific traffic rules. In this paradigm, modeling of the future serves merely as an intermediate latent variable en route to the final action, **lacking an explicit, independent physical supervision signal**. As long as the final values matches the ground-truth label, the model incurs no penalty even if its **internal understanding of environmental evolution is physically implausible**. This fails to quantify the model's understanding of the world.
> > >
> > > In contrast, DeepSight is dedicated to constructing a true **Generative World Model** that explicitly learns the state transition dynamics between current and future environments: $P(P_t, T_{cot}, \mathbf{F} \mid X)$. This forces the model to internally simulate **what the world will look like** before deciding **how to act.** Such explicit modeling of future states establishes a causal link between environmental evolution and trajectory planning. We compel the model to explicitly reconstruct future latent semantic features via learnable **World Queries ($Q_{world}$)** prior to action planning. Crucially, this reconstruction is rigorously supervised by a dedicated **World State Prediction Loss**:
> > >
> > > $$ L_{world} = MSE(F_{predicted},F_{GT})$$
> > >
> > >
> > >
> > > This mechanism forces the model to accurately capture the physical laws governing **environmental evolution**, rather than simply **fitting action labels**.
> > >
> > > We summarize the comparisons across multiple dimensions in the table below.
> > >
> > > | Dimension | VLM-based E2E Methods | DeepSight |
> > > | :--- | :--- | :--- |
> > > | **1. Representation** | Predicts Next Token: Essentially a **classification task** that fails to capture continuous spatiotemporal nuances. | Predicts vectors aligned with semantic features: Essentially a **regression task** that preserves fine-grained geometric details and motion dynamics. |
> > > | **2. Data Efficiency** | Label-Dependent: Relies entirely on expensive expert annotations. | Self-Supervised Friendly: Requires only **video sequences**. Enables learning physical laws from **massive unlabeled data**. |
> > > | **3. Reasoning** | Direct Mapping: Maps current state directly to future actions, lacking any intermediate verification step. | Internal Simulation: Enables **verifiable trajectory planning** by leveraging internally simulated future states. |
> > > ***
> > > Thank you again for your time and insightful comments. We hope our responses have effectively addressed your concerns and we are more than happy to include all these discussions in the camera-ready version of this work.
> > > ***
> > >
> > > Assran et al., 2025: V-JEPA 2: Self-Supervised Video Models Enable Understanding, Prediction and Planning. In arXiv
> > >
> > > Zeng et al., 2025: FutureSightDrive: Thinking Visually with Spatio-Temporal CoT for autonomous Driving. In NeurIPS
> > >
> > > Ye et al., 2026: World Action Models are Zero-shot Policies. In arXiv

---

### Official Review · Reviewer_ZQnF · 2026-03-10

**Soundness:** 3
**Presentation:** 3
**Significance:** 3
**Originality:** 2
**Overall Recommendation:** 4
**Confidence:** 4

**Summary:**

This paper proposes DeepSight, an end-to-end autonomous driving framework that integrates a latent BEV world model with adaptive reasoning. The method predicts future multi-frame latent semantic features in BEV space to model long-horizon world dynamics and improve trajectory planning. In addition, an Adaptive Chain-of-Thought (CoT) mechanism is introduced to selectively activate reasoning based on scene complexity, enhancing decision-making in long-tail driving scenarios. Experiments on the Bench2Drive closed-loop benchmark demonstrate significant improvements over prior methods in both driving performance and efficiency.

**Compliance With Llm Reviewing Policy:**

Affirmed.

**Final Justification:**

The authors’ clarifications have fully addressed my previous concerns and resolved my confusion. Thus I adjust the score to "weak accept".

**Key Questions For Authors:**

**Questions:**

- As mentioned in the weaknesses, the experimental evaluation focuses primarily on the Bench2Drive benchmark. Could the authors comment on how well the proposed method generalizes to other autonomous driving datasets (e.g., navsim)? Additional experiments or discussion on cross-dataset generalization would strengthen the paper.

- The ablation study suggests that most of the performance improvement comes from the world modeling component, while the Adaptive CoT module provides a relatively smaller gain. It would be helpful if the authors could further analyze in which types of scenarios the reasoning module provides the most benefit.

- Could the authors clarify how “long-horizon” is defined in this work? In particular, what temporal range can the model reliably predict, and how does performance change as the horizon increases?

**Limitations:**

Yes.

**Strengths And Weaknesses:**

**Strengths:**

- Propose a latent BEV-based world modeling approach that predicts future semantic features for multiple frames in parallel, enabling the model to reason about longer temporal horizons compared to autoregressive prediction methods. This design improves the model’s ability to capture scene dynamics and supports more reliable trajectory planning.

- Adaptive CoT enables the system to incorporate external knowledge when needed while avoiding unnecessary computational overhead in simple scenarios.


**Weaknesses:**

- The paper brings together several existing techniques, including BEV-based world modeling, latent feature prediction, and reasoning mechanisms. While the overall system design is well organized, the individual components are largely based on previously explored ideas.

- The experiments are conducted primarily on Bench2Drive, which is a simulator-based benchmark. Additional evaluation on other datasets or benchmarks (e.g., navsim [1]) would make the paper more convincing.

- The ablation results shown in Tab. 5 suggests that most of the performance gain comes from the world modeling component, while the adaptive CoT module contributes relatively smaller improvements.

[1] Dauner, Daniel, et al. "Navsim: Data-driven non-reactive autonomous vehicle simulation and benchmarking." Advances in Neural Information Processing Systems 37 (2024): 28706-28719.

---

> ### Author Rebuttal · Authors · 2026-03-30
>
> We sincerely thank the reviewer for their positive and insightful feedback. We are particularly encouraged by your recognition of the novelty in our proposed **_BEV-based world modeling_** approach. You have also highlighted several issues requiring clarification. Below, we provide detailed responses to these points, accompanied by supplementary experiments.
> ***
> **_Q1: While the overall system design is well organized, the individual components are largely based on previously explored ideas._**
>
> A1: We sincerely thank the reviewer for acknowledging the well-organized system design of DeepSight. However, we respectfully clarify that DeepSight is not a simple stacking of existing modules, but rather a comprehensive innovation driven by deep insights into the specific challenges.
> Unlike existing methods that merely add a BEV module, we designed a **native VLLM architecture** capable of predicting latent semantic features for **consecutive future multi-frames**. Simply replacing pixels with latents is insufficient. We introduced specific architectural improvements **tailored for multi-frame long-term temporal modeling and parallel inference**. We did not blindly pile on CoT, which often incurs prohibitive latency. Recognizing the **trade-off between the logical strengths of CoT and its computational costs**, we devised an efficient adaptive mechanism. In summary, the **SOTA** results achieved on the Bench2Drive and nuScenes (The question mentioned in Q2) are not the result of simple combinations.
> ***
> **_Q2: Could the authors comment on how well the proposed method generalizes to other autonomous driving datasets?_**
>
> A2: We fully agree that evaluation on real data is critical for verifying the practical applicability of our approach. We also experimented with the navsim dataset. However, due to its **discontinuous sampling**, only **39000** out of **85000** samples were valid. To this end, we have extended our evaluation to the **nuScenes** dataset. For a fair comparison, FSDrive was reproduced using the same base model and experimental settings as ours.
>
> |Method|L2 (1s) ↓|L2 (2s) ↓|L2 (3s) ↓|L2 (Avg) ↓|Col (1s) ↓|Col (2s) ↓|Col (3s) ↓|Col (Avg)↓|
> |-|:-:|:-:|:-:|:-:|:-:|:-:|:-:|:-:|
> |LAW|0.26|0.57|1.01|0.61|0.14|0.21|0.54|0.30|
> |World4Drive|0.23|0.47|0.81|0.50|0.02|0.12|0.33|0.16|
> |FSDrive*|0.27|0.33|0.56|0.35|0.07|0.10|0.24|0.14|
> |**DeepSight**|**0.16**|**0.31**|**0.52**|**0.33**|**0.02**|**0.07**|0.27|**0.12**|
>
> Both open-loop and closed-loop experiments confirm that DeepSight outperforms **LAW and FSDrive**.
>
> ***
> **_Q3: Analysis of the scenario types where the reasoning module provides the most benefit is needed._**
>
> A3: We thank the reviewer for this insightful observation. Although the Adaptive CoT module contributes a marginal gain to the DS, it plays a disproportionately critical role in specific long-tail scenarios. Crucially, we did not blindly stack CoT modules. Instead, **recognizing the limitations of world modeling in certain contexts** and carefully **weighing the logical reasoning strengths of CoT against its computational latency**, we devised an efficient **adaptive mechanism** to holistically refine our algorithmic framework. Statistical analysis of our closed-loop evaluation across 220 routes reveals that this reasoning mechanism was triggered in less than **30%** of frames, ensuring efficiency.
>
> Our visualizations highlight three distinct cases necessitating external knowledge reasoning, as illustrated in Figure 3. For rare events underrepresented in the training distribution, such as encountering fire trucks and construction warning signs, the world model may lack sufficient priors. In these instances, the Adaptive CoT module effectively bridges this gap by leveraging external knowledge to guide decision-making. World Modeling serves as the primary innovation of our work by enabling precise long-horizon trajectory prediction; in contrast, **Adaptive CoT** represents our strategic effort to enhance **practical applicability and robustness** in complex scenarios.
> ***
> **_Q4: Could the authors clarify how long-horizon is defined in this work?_**
>
> A4: The definition of a long horizon in our work is relative to prior studies, which were typically limited to predicting only 0.5 seconds into the future. In contrast, DeepSight can predict up to **2 seconds** ahead, encompassing a total of 5 frames of DINO features, including the current frame. Visualizations are available at: /https://anonymous.4open.science/r/test-C7FB.
>
> As observed, the accuracy of the predicted semantic features gradually declines as the prediction horizon extends. Specifically, **error accumulation** becomes evident in the latter half of the **2-second prediction window**. Consequently, we determined that a 2-second horizon represents the optimal trade-off for achieving high-quality action generation. In future work, we aim to extend our framework to support reliable predictions over even longer time ranges.

---

> > ### Author Rebuttal · Reviewer_ZQnF · 2026-04-01
> >
> > The additional experimental results provide strong evidence for the effectiveness of the proposed DeepSight. The authors’ clarifications have fully addressed my previous concerns and resolved my confusion. Overall, the revision has substantially improved both the technical soundness and clarity of the paper. Accordingly, I will increase my score.

---

> > > ### Author Response · Authors · 2026-04-06
> > >
> > > Thank you for your thoughtful consideration and for recognizing the value our work may bring to the community. We appreciate your assessment and support.

---

### Official Review · Reviewer_dRoZ · 2026-03-12

**Soundness:** 2
**Presentation:** 3
**Significance:** 2
**Originality:** 3
**Overall Recommendation:** 5
**Confidence:** 4

**Summary:**

The paper introduces DeepSight, a VLM for end-to-end driving that performs latent world modeling, adaptive chain-of-thought reasoning, and trajectory planning. The proposed world modeling predicts latent semantic features for future time steps. Furthermore, the model is trained to dynamically activate CoT reasoning based on the features/complexity of a driving scenario.

**Compliance With Llm Reviewing Policy:**

Affirmed.

**Final Justification:**

The authors resolved all my concerns (weaknesses) by performing an additional experiment, revising parts of the motivation and adding  results of related methods.

**Key Questions For Authors:**

Do the authors have an hypothesis why CoT improves the results of DeepSight but worsens the results for related methods like DriveLM (Sima et al., 2024) or Poutine (Rowe et al., 2025)?

Sima et al., 2024: DriveLM: Driving with Graph Visual Question Answering. In ECCV

Rowe et al. 2025: Poutine: Vision-Language-Trajectory Pre-Training and Reinforcement Learning Post-Training Enable End-to-End Autonomous Driving. In arXiv

**Limitations:**

yes

**Strengths And Weaknesses:**

Strengths:
- Using pre-trained DINOv3 models to generate rich latent representations of future world states is well motivated by the success of self-supervised vision encoders in other domains. Ablating further encoders (e.g., JEPA (Assran et al., 2023) or VICReg (Bardes et al., 2022) models) would further improve the paper.
- The adaptive chain-of-thought mechanism is well designed for the varying complexity of driving scenarios (i.e., nominal vs. long-tail scenarios).
- Overall, the paper is well written and well structured.

Weaknesses:
- Table 1 and 2 are missing the results of SimLingo (Renz et al., 2025), which is mentioned in the related work section. The results of DeepSight are strong but the gap to other VLMs is smaller than reported.
- The evaluation is only performed on one dataset, which has a large sim-to-real domain-gap. An additional evaluation on a dataset like WOD-E2E (Xu et al., 2025) would fit the focus on long-tail driving events and significantly improve the paper.
- Minor flaw in the motivation (line 33 right): Humans are usually not viewed as being particularly good at long-horizon motion modeling but rather at predicting coarser intentions.
- Minor formatting issue: The running title at the top of each page is still from the template.
- Minor formatting issue in line 328 left: Subsection title at the bottom of the page.

Assran et al., 2023: Self-Supervised Learning from Images with a Joint-Embedding Predictive Architecture. In CVPR

Bardes et al., 2022: VICReg: Variance-Invariance-Covariance Regularization for Self-Supervised Learning. In ICLR

Xu et al., 2025: WOD-E2E: Waymo Open Dataset for End-to-End Driving in Challenging Long-tail Scenarios. In arXiv

---

> ### Author Rebuttal · Authors · 2026-03-30
>
> We sincerely appreciate your insightful and professional comments. We are particularly encouraged by your recognition of our core contribution: **_Using pre-trained DINOv3 models to generate rich latent representations of future world states_**. We also extend our gratitude for your high praise regarding the clarity of our manuscript.
> Beyond acknowledging the strengths of our work, we have carefully considered your constructive feedback and questions, conducting additional comprehensive experiments to address them thoroughly.
> ***
> **_Q1: Table 1 and 2 are missing the results of SimLingo._**
>
> A1: We thank the reviewer for pointing out the absence of SimLingo results in our tables. The primary reason SimLingo was excluded from Tables 1 and 2 lies in the inconsistency of **the expert data used for training**: SimLingo is trained on demonstrations generated by **PDM-Lite**, whereas DeepSight and Orion strictly adhere to the **official Bench2Drive** protocol, which utilizes models trained on **Think2Drive** demonstrations. This discrepancy in training sources prevents a direct comparison. In future work, we plan to evaluate DeepSight within the PDM-Lite setting as well, to provide a comprehensive cross-benchmark analysis.
> ***
> **_Q2: The evaluation is only performed on one dataset, which has a large sim-to-real domain-gap._**
>
> A2: We fully agree that evaluation on real data is critical for verifying the practical applicability of our approach. To this end, we have extended our evaluation to the **nuScenes** dataset, a widely adopted benchmark for real-world autonomous driving. We adhered to the standard evaluation protocols employed in recent works, utilizing metrics such as **L2** and **Collision**. For a fair comparison, FSDrive was reproduced using the same base model and experimental settings as ours.
>
> | Method | L2 (1s) ↓ | L2 (2s) ↓ | L2 (3s) ↓ | L2 (Avg) ↓ | Col (1s) ↓ | Col (2s) ↓ | Col (3s) ↓ | Col (Avg) ↓ |
> |-|:-:|:-:|:-:|:-:|:-:|:-:|:-:|:-:|
> |ST-P3|1.33|2.11|2.90|2.11|0.23|0.62|1.27|0.71|
> |UniAD|0.48|0.96|1.65|1.03|0.05|0.17|0.71|0.31|
> |VAD|0.41|0.70|1.05|0.72|0.07|0.18|0.43|0.23|
> |LAW|0.26|0.57|1.01|0.61|0.14|0.21|0.54|0.30|
> |World4Drive|0.23|0.47|0.81|0.50|0.02|0.12|0.33|0.16|
> |FSDrive*|0.27|0.33|0.56|0.35|0.07|0.10|0.24|0.14|
> |**DeepSight**|**0.16**|**0.31**|**0.52**|**0.33**|**0.02**|**0.07**|0.27|**0.12**|
> ***
> **_Q3: Minor formatting issue._**
>
> A3: Thanks for the catch. We will correct these typos and conduct a complete check in the camera-ready version.
> ***
> **_Q4: Minor flaw in the motivation._**
>
> A4: We thank the reviewer for this insightful correction. We agree that humans excel at predicting coarse intentions rather than precise long-term trajectories. It is important to clarify that our definition of **long-horizon** refers to the 2-second sequence required for autonomous driving planning, as opposed to single-frame or 0.5-second predictions. At this timescale, predicting future semantic states is both feasible and necessary. In fact, aligning with the reviewer's insight, we are not performing low-level pixel motion prediction, but rather **semantic-level state prediction**. This approach explicitly mimics the human capability to predict **intentions** rather than precise physical trajectories. In the camera-ready version, we will modify the sentence to better reflect this nuance.
> ***
> **_Q5: Do the authors have an hypothesis why CoT improves the results of DeepSight but worsens the results for related methods like DriveLM or Poutine?_**
>
> A5: We thank the reviewer for this insightful question. We posit that the divergence in CoT performance between DeepSight and methods like DriveLM or Poutine stems primarily from **scenario complexity**.
> Regarding the performance degradation observed in DriveLM, we attribute this to error propagation. As evidenced by Table 2 in the DriveLM study, the **Chain** mode underperforms even the **None** baseline. We hypothesize that the nuScenes dataset, which consists predominantly of nominal scenarios, does not necessitate such rigid multi-step reasoning. Similarly, for Poutine, **the generation of lengthy textual reasoning traces likely introduces cumulative errors** or **dilutes the model's attention**, preventing further performance gains. These observations directly motivate our design of Adaptive CoT. We contend that not every scenario warrants deep, multi-step reasoning; imposing it universally not only compromises efficiency but also degrades performance due to error accumulation. In contrast, DeepSight dynamically modulates its reasoning depth based on scene complexity. This approach effectively avoids unnecessary error propagation while preserving the model's capacity for logical inference in critical situations, thereby achieving an optimal balance between robustness and efficiency.

---

> > ### Author Rebuttal · Reviewer_dRoZ · 2026-04-03
> >
> > A2 - A5 resolve the corresponding weaknesses. Especially the results on nuScenes are strong and the argument that lengthy textual reasoning likely introduces cumulative errors is convincing. Regarding A1, I suggest still adding the SimLingo results with a footnote explaining the different setup or by adding a new column to specify PDM-Lite vs. Think2Drive demonstrations. Overall, I'm considering to raise my score.

---

> > > ### Author Response · Authors · 2026-04-04
> > >
> > > We sincerely thank the reviewer for the positive feedback and the encouraging comment. Therefore, in the camera-ready version, we will add **a new column** to the relevant table to explicitly specify the expert data source, clearly distinguishing between PDM-Lite and Think2Drive demonstrations. Additionally, we will include the SimLingo results as suggested.
> > >
> > > Thank you again for your constructive guidance, which has significantly improved the quality of our paper.

---

### Official Review · Reviewer_YSCs · 2026-03-12

**Soundness:** 3
**Presentation:** 2
**Significance:** 3
**Originality:** 2
**Overall Recommendation:** 3
**Confidence:** 5

**Summary:**

The authors recognize the limitation sof existing world modeling and visual reasoning in terms of the forms of visual representation and long-horizon prediction. Specifically, they argue that prioritizing image texture over semantic information is not appropriate, and that, for end-to-end (E2E) autonomous driving, it is vital to predict the long-term future and model surrounding agents, not just the front-view images. The authors therefore propose a VLM-based E2E driving framework, named DeepSight, with two major contributions. First, it integrates a driving world model that performs parallel prediction of the latent semantic features using DINO v3 as the supervisor for multiple consecutive future frames in the BEV space, which helps enable long-term (_i.e,_ two seconds in this paper) future state prediction. Meanwhile, it comes with an adaptive Chain-of-Thought mechanism that conditionally activates textual reasoning for long-tail scenarios. Their experiments showcase that DeepSight achieves state-of-the-art driving score and success rate compared to existing models, including AutoVLA and ORION.

**Compliance With Llm Reviewing Policy:**

Affirmed.

**Final Justification:**

I am impressed by their self-confidence, as the authors do not hesitate to use terms like "**comprehensive innovation**" in their rebuttal, while admitting that one of their listed contributions _**employs**_ the DINO features. In the rebuttal, I would expect the authors to support their claims, such as "_...fails to leverage the VLM's inherent causal reasoning capabilities while incurring unnecessary training and inference overhead_", with solid figures from comparative studies and quantitative analyses, rather than unfounded accusations. All in all, I appreciate the authors' efforts in addressing my concerns, but I will keep my original rating.

**Key Questions For Authors:**

1. As mentioned above, how exactly do you incorporate the world features into the LLM, and how does a different design of such integration affect the performance?
2. As mentioned above, how does the method compare to LAW and FSDrive? I believe these are the most directly related latent-world-model approaches from recent years.
3. Despite recognizing a trade-off in comfort scores, can you provide a more in-depth analysis of what types of uncomfortable behaviors the model might exhibit? I am suspecting whether a high efficiency score is achieved by cheating by aggressively speeding, which degrades the practical applicability of such a method.
4. Please elaborate more on how you calibrate the $\lambda$'s and how sensitive the performance is to these hyperparameters.

**Limitations:**

The authors briefly highlight the computational costs as a major limitation. However, several other potential limitations are not discussed. For example, I would expect a more detailed elaboration on the trade-off between comfort scores and how reasonable such a trade-off is in real-world practice. Meanwhile, I would be more than happy to know whether the CoT annotation pipeline can be easily scaled up and how it might potentially fail.

**Strengths And Weaknesses:**

I am surprised by the astonishing empirical performance presented by this paper. The results show a substantial $7.39$ improvement in driving score and an $\approx 14\%$ improvement in success rate over the existing AutoVLA on Bench2Drive. Table 2 further shows that the proposed method outperforms the competitor by a large margin on the benchmark. Meanwhile, the authors conduct a well-designed ablation study that makes the results more convincing to me. The finding that VAE-based world modeling actually degrades when extended to multi-frame prediction, while latent semantic features improve the driving score, is a compelling justification for the design choice. The efficiency analysis (Table 6), showing only $3.57\%$ additional latency for the world model, is also valuable. Finally, let us not forget that the authors claim they will contribute $1.3$M in adaptive CoT annotations for `Bench2Drive`, which is valuable to the open-source community.

Nevertheless, I believe the authors need to address a few concerns to improve the overall quality of this story. First and most importantly, despite all the efforts, the method itself is not novel. The two main components, the latent BEV feature prediction for world modeling and the text CoT for reasoning, are both well-explored ideas on their own, if I remember correctly. Latent world models for driving have been proposed in LAW, presented at ICLR 2025, and in BEVWorld and World4Drive, presented at ICCV 2025. Adaptive or conditional CoT for driving is implemented in SimLingo, AdaThinkDrive, and CoT-VLA, among others. The paper's contribution is primarily in combining these ideas within a unified VLM framework and demonstrating strong Bench2Drive results, but the architectural novelty is limited. The parallel multi-frame prediction via learnable queries is relatively straightforward given existing transformer-based decoding paradigms. Therefore, I think this paper qualifies on the basis of its level of effort, but not its novelty.

That said, I believe the paper is missing critical comparisons with key concurrent work. The paper does not compare with several highly relevant recent methods: FSDrive/FutureSightDrive (NeurIPS 2025 spotlight, which also does VLM-based world modeling + CoT for driving), LAW (ICLR 2025, latent world model for driving), DriveVLA-W0, or Centaur. Given the rapidly moving field, the comparison set in Table 1 is somewhat incomplete for the latest VLM-based approaches. I think this is something that authors can complete in their next step. It is essential for the authors to clarify how they distinguish their work from these concurrent studies and how each novel contribution advances the task itself.

Moreover, the authors claim that the world models predict future latent features, which will be used in trajectory generation. But I might need more details to understand how the LLM actually uses these latent features. Are they simply concatenated into the sequence of tokens, or is there a use of attention to incorporate these as conditions? Equation 6 now looks more like a high-level formulation but omits some critical implementation details.

---

> ### Author Rebuttal · Authors · 2026-03-30
>
> We really appreciate the reviewer for the detailed and constructive feedback. We believe our response has addressed most of your concerns and we respectfully ask **if you're comfortable to re-consider the rating in light of our efforts and dedication**.
> ***
> **_Q1: How are world features integrated into the LLM?_**
>
> A1: Instead of injecting external features, we leverage the MLLM's native hidden states. Jointly predicting world features and text decomposes the task into: (1) World prediction, (2) CoT reasoning, and (3) Waypoint prediction.
>
> $$
> p(P_t, T_{cot}, F \mid X) =
> \\underset{\\text{World Prediction}}{\\underline{p(F \mid X)}} \\cdot
> \\underset{\\text{Reasoning}}{\\underline{p(T_{cot} \mid X, F)}} \\cdot
> \\underset{\\text{Planning}}{\\underline{p(P_t \mid X, F, T_{cot})}}
> $$
>
> The LLM's autoregressive prediction ensures that the output of each subsequent module is based on the **attention** computations derived from the preceding module. DeepSight first generates latent world features $F$, grounding subsequent CoT in anticipated future states rather than just current observations. Finally, trajectory tokens, $P_t$ are predicted based on the full context ($X, F, T_{\text{cot}}$). Furthermore, Table 3 presents ablation studies on the impact of prediction objectives and horizons.
> ***
> **_Q2: How does the method compare to LAW and FSDrive?_**
>
> A2: DeepSight's key advantage over SOTA methods is its effective **use of future understanding for planning**. Unlike LAW, which employs the world model head merely as an auxiliary **training task** and typically discards these outputs during inference, or FSDrive, limited by the inefficiency and poor semantics of generating **384 VAE tokens per image**, DeepSight aligns DINO features in **hidden layers** to enable efficient, prediction-conditioned trajectory planning. Comparative visualizations are available at: /https://anonymous.4open.science/r/test-C7FB. Detailed table of differences is provided in our response to **Reviewer GAZN, Q1** due to space limitations.
>
> We reproduced FSDrive's missing closed-loop results on Bench2Drive, as shown in Table 3 (Row 1)
>
> |**METHOD**|**RC**↑ | **IS**↑| **DS**↑ |
> |-|-|-|-|
> |FSdrive|47.56|0.64|27.75|
> |**DeepSight**|**95.95**|**0.89**|**86.57**|
>
> We also compared our method with LAW and FSDrive on **nuScenes** dataset.
>
> | Method | L2 (1s) ↓ | L2 (2s) ↓ | L2 (3s) ↓ | L2 (Avg) ↓ | Col (1s) ↓ | Col (2s) ↓ | Col (3s) ↓ | Col (Avg) ↓ |
> |:-|:-:|:-:|:-:|:-:|:-:|:-:|:-:|:-:|
> |LAW|0.26|0.57|1.01|0.61|0.14|0.21|0.54|0.30|
> |World4Drive|0.23|0.47|0.81|0.50|0.02|0.12|0.33|0.16|
> |FSDrive*|0.27|0.33|0.56|0.35|0.07|0.10|0.24|0.14|
> |**DeepSight**|**0.16**|**0.31**|**0.52**|**0.33**|**0.02**|**0.07**|0.27|**0.12**|
>
> For a fair comparison, FSDrive was reproduced using the same base model and experimental settings as ours. Both **open-loop** and **closed-loop** experiments confirm that DeepSight **outperforms** LAW and FSDrive.
> ***
> **_Q3: A deeper analysis of uncomfortable behaviors and the trade-off between comfort scores._**
>
> A3:  Our model prioritizes safety over comfort in critical scenarios, such as **emergency stops**, leading to occasional deceleration spikes. This is compounded by **inherent noise and unstable segments** in the expert demonstration training data. Notably, modest comfort scores are a broader trend in Bench2Drive research, for instance, Orion achieved only 17.38. We clarify that our high efficiency stems from proactive long-horizon planning, not aggressive speeding. In Bench2Drive, **speeding incurs severe penalties** that would drastically lower the DS. Yet, our method achieved the highest **86.23** DS and **71.36** SR, significantly outperforming baselines. Our efficiency gains result from reducing **unnecessary stops** through superior world modeling, strictly adhering to official configurations without external data.
> ***
> **_Q4: Please elaborate on the calibration of $\lambda$ values._**
>
> A4: Since both trajectory waypoints and CoT reasoning are encoded as text tokens, they exhibit similar gradient magnitudes and convergence behaviors. We calibrate both terms using a shared hyperparameter $\lambda_{cot} = \lambda_{traj} = 1.0$. To evaluate hyperparameter sensitivity on Dev 10 set, we varied $\lambda_{world} \in \{0.5, 1.0, 2.0\}$. Results are summarized below:
>
> |$\lambda_{world}$|RC↑|IS↑|DS↑|
> |-|-|-|-|
> |0.5|88.23|0.89|78.93|
> |**1.0**|**95.95**|**0.89**|**86.57**|
> |2.0|89.12|0.89|79.82|
>
> Interestingly, our experiments demonstrate a non-monotonic relationship where performance peaks and then drops as
> λ increases. We attribute this phenomenon to the inherent ambiguity of world modeling. While an excessively high weight can degrade final performance, it still outperforms scenarios with minimal world modeling.
> ***
> **_Q5: The CoT annotation pipeline's details._**
>
> A5: Our automated CoT annotation pipeline leverages LLM distillation for exceptional scalability, with format and logic checks ensuring high quality.

---

> > ### Author Rebuttal · Reviewer_YSCs · 2026-04-05
> >
> > My sincere appreciation to the authors for taking my questions seriously. I believe several of my concerns have been addressed.
> >
> > However, I have several remaining concerns. First, the significance and novelty of the proposed method remain my primary concern. I believe that the two main contributions in this work, including latent BEV world modeling and adaptive CoT, are well-explored individually. The rebuttal demonstrates strong empirical results against LAW and FSDrive, which I appreciate, but it does not articulate what is architecturally novel about DeepSight beyond combining these ideas in a VLM framework. Meanwhile, the rebuttal adds LAW and FSDrive, which is helpful. But my review also asked about Centaur (which does test-time training for driving) and DriveVLA-W0. These remain unaddressed. Are these comparative results available? Also, you mention, quote, "_format and logic checks ensuring high quality_," but my review asked about failure modes and scalability challenges. What proportion of annotations require correction after autolabeling? What types of scenarios does the pipeline struggle with? Given that you are contributing 1.3M annotations to the community, I am convinced that a more rigorous quality analysis would strengthen the claim.

---

> > > ### Author Response · Authors · 2026-04-07
> > >
> > > We sincerely appreciate your thoughtful review and recognition of DeepSight. Below, we clarify our contributions.
> > > ***
> > > _**Q1: DeepSight's novelty.**_
> > >
> > > A1: DeepSight is not a trivial module concatenation but **a comprehensive innovation** driven by insights into autonomous driving challenges and efficiency needs. Our design ensures: (a) intrinsic module effectiveness, (b) tight inter-module synergy, and (c) overall system efficiency. DeepSight's novelty lies in three core innovations.
> > >
> > > **Latent Feature vs. Pixel Reconstruction**: Unlike traditional world models that rely on VAEs for pixel-level reconstruction (e.g., FSDrive, DriveVLA-W0), DeepSight innovatively employs **DINO features** as its latent representation. This design shift enables the model to focus on capturing key semantic information of the environment.
> > >
> > > **Long-Horizon Parallel Modeling**: We introduce **specific architectural improvements tailored for multi-frame long-horizon modeling and parallel inference**. Traditional AR methods suffer from low efficiency, training-inference inconsistency, and severe error accumulation when handling long-term tasks. Our parallel decoding mechanism overcomes these limitations, enabling efficient and accurate prediction of long-range trajectories without step-by-step generation.
> > >
> > > **Adaptive CoT**: We posit that the world model inherently possesses reasoning capabilities for routine scenarios such as tracking vehicle dynamics. However, for long-tail corner cases, such as yielding to emergency vehicles, leveraging the extensive world knowledge of LLMs for deep reasoning is crucial. Consequently, we designed an Adaptive CoT mechanism that fosters tight synergy between the world model and the LLM while minimizing additional computational overhead.
> > >
> > > Our SOTA results on **Bench2Drive and nuScenes** directly stem from these architectural innovations, not simple module stacking.
> > > ***
> > > _**Q2: How does DeepSight compare to Centaur and DriveVLA-W0?**_
> > >
> > > A2: Centaur does **not employ a generative world model** for planning. Instead, it relies on Test-Time Training to minimize uncertainty over a fixed, pre-defined trajectory vocabulary. Specifically, it samples candidates from this static set, clusters them, and updates model parameters via gradient descent to sharpen the score distribution. In contrast, DeepSight learns the underlying physical dynamics of the environment through world modeling.
> > >
> > > DriveVLA-W0 incorporates world modeling via two distinct pathways: (1) Autoregressive token prediction, which mirrors FSDrive and even utilizes the identical MoVQGAN encoder; and (2) a Diffusion-based approach to predict latent features of future images. The AR pathway not only compromises efficiency but also exacerbates error accumulation in long-horizon tasks, compelling DriveVLA-W0 to restrict its prediction to merely **the next single frame**.
> > >
> > > The Diffusion pathway requires introducing additional architectural components, effectively decoupling the world model from the trajectory planner. We argue that this separation **fails to leverage the VLM's inherent causal reasoning capabilities while incurring unnecessary training and inference overhead**.
> > >
> > > Centaur and DriveVLA-W0 only report on NAVSIM, an open-loop benchmark whose discontinuous sampling (~39k/85k valid frames) disrupts the temporal consistency essential for long-horizon modeling.
> > >
> > > In conclusion, extensive comparisons with existing SOTA methods underscore our fundamental innovations and distinctiveness, which further substantiate our response to **Q1**. We sincerely appreciate your reference to these papers and will incorporate this comprehensive discussion into the camera-ready version.
> > > ***
> > > _**Q3: The CoT annotation pipeline's details.**_
> > >
> > > A3: As detailed in our Appendix, we examined and filtered two types of errors: **when the model judged a scenario to be simple but still provided some reasoning, and when it judged a scenario to be complex but did not provide any reasoning**. Subsequently, an analysis of 5k failure cases in complex scenarios identified two predominant failure modes. **Adverse Weather Conditions**: In heavy rain or fog, the interplay between weather effects and traffic participants occasionally led to generation failures. **Occlusion and Density**: Scenarios with severe occlusion or high densities of traffic participants sometimes impaired the model's judgment.
> > >
> > > According to our statistics, **4.72%** of the samples initially generated using the open-source Qwen3VL-235B required correction. We anticipate that employing more powerful commercial models would yield even lower failure rates.
> > > A more significant challenge lies in detecting plausible hallucinations, which are inherently difficult to identify. To further enhance data quality, future work could **incorporate more sophisticated agent systems**, such as ReAct, Reflexion, Self-Correct, or RLEF.
> > > ***
> > > Thank you again for your follow-up questions. We hope these clarifications resolve your concerns.

---

### Decision · Program_Chairs · 2026-04-30

**Decision:**

Accept (regular)

**Comment:**

DeepSight combines parallel latent world-state prediction (DINOv3-supervised) with adaptive Chain-of-Thought reasoning in a VLM-based end-to-end self-driving, achieving strong Bench2Drive performance with minimal inference overhead (~3.6%). Reviewer dRoZ (Accept) finds the approach well-designed and concerns fully resolved. Reviewer YSCs (Weak Reject) acknowledges strong empirical results but questions architectural novelty, noting the work combines existing ideas (latent BEV prediction, adaptive CoT) and misses comparisons to concurrent methods (SimLingo, Centaur, DriveVLA-W0). The 1.3M CoT annotation contribution and the efficiency of selective reasoning activation are genuine strengths. The novelty concern has merit but the empirical contribution and practical design warrant acceptance.